# Exosomal Osteoclast-Derived miRNA in Rheumatoid Arthritis: From Their Pathogenesis in Bone Erosion to New Therapeutic Approaches

**DOI:** 10.3390/ijms25031506

**Published:** 2024-01-25

**Authors:** Sandra Pascual-García, Pascual Martínez-Peinado, Carolina Pujalte-Satorre, Alicia Navarro-Sempere, Jorge Esteve-Girbés, Ana B. López-Jaén, Juan Javaloyes-Antón, Raúl Cobo-Velacoracho, Francisco J. Navarro-Blasco, José M. Sempere-Ortells

**Affiliations:** 1Department of Biotechnology, University of Alicante, 03690 San Vicente del Raspeig, Spain; 2Department of Legal Studies of the State, University of Alicante, 03690 San Vicente del Raspeig, Spain; 3Department of Physics, Systems Engineering and Signal Theory, University of Alicante, 03690 San Vicente del Raspeig, Spain; 4Rheumatology Unit, University General Hospital of Elche, 03203 Elche, Spain

**Keywords:** rheumatoid arthritis, osteoclast, exosomes, miRNAs, osteoblast

## Abstract

Rheumatoid arthritis (RA) is an autoimmune disease that causes inflammation, pain, and ultimately, bone erosion of the joints. The causes of this disease are multifactorial, including genetic factors, such as the presence of the human leukocyte antigen (HLA)-DRB1*04 variant, alterations in the microbiota, or immune factors including increased cytotoxic T lymphocytes (CTLs), neutrophils, or elevated M1 macrophages which, taken together, produce high levels of pro-inflammatory cytokines. In this review, we focused on the function exerted by osteoclasts on osteoblasts and other osteoclasts by means of the release of exosomal microRNAs (miRNAs). Based on a thorough revision, we classified these molecules into three categories according to their function: osteoclast inhibitors (miR-23a, miR-29b, and miR-214), osteoblast inhibitors (miR-22-3p, miR-26a, miR-27a, miR-29a, miR-125b, and miR-146a), and osteoblast enhancers (miR-20a, miR-34a, miR-96, miR-106a, miR-142, miR-199a, miR-324, and miR-486b). Finally, we analyzed potential therapeutic targets of these exosomal miRNAs, such as the use of antagomiRs, blockmiRs, agomiRs and competitive endogenous RNAs (ceRNAs), which are already being tested in murine and ex vivo models of RA. These strategies might have an important role in reestablishing the regulation of osteoclast and osteoblast differentiation making progress in the development of personalized medicine.

## 1. Introduction

### 1.1. Introduction to Rheumatoid Arthritis

Rheumatoid arthritis (RA) is an autoimmune condition that primarily affects the joints, as well as connective tissue, muscles, and tendons. This disease has a prevalence of 0.54% in Europe [1], typically appearing around the age of 45 and triplicating its incidence in women in comparison with men [2,3].

In this disease, there is an inflammation of the synovial membrane that covers the joints (referred to as synovitis) and the development of invasive synovial tissue known as pannus, which, over time, leads to the degradation of cartilage, bone, and the joint itself [4]. The pannus is abnormally highly vascularized, promoting a pro-inflammatory environment that contributes to greater joint degradation [5].

RA presents four stages based on symptom intensity and exacerbation: early, moderate, severe, and end-stage. The early phase is characterized by joint inflammation and stiffness, possibly accompanied by symptoms like fever, fatigue, and loss of appetite. In the moderate stage, joint cartilage is inflamed, leading to pain and reduced mobility. The severe stage involves bone inflammation, worsening the previously mentioned symptoms and potentially causing bone erosion. Patients at this stage may also experience muscle weakness or atrophy. Ultimately, in the end-stage, joints undergo complete erosion, losing their ability to facilitate bone mobility. Additionally, due to these changes, bone fusion, known as ankylosis, can occur, resulting in a definitive loss of function [6]. However, not all RA patients experience all four stages, as treatment with disease-modifying antirheumatic drugs (DMARDs) aims for remission, reducing symptoms to enable a normal life [7].

Regarding its causes, there is no single reason for this disease; rather, it results from a combination of genetic, epigenetic, microbiome-related, environmental, and immunological factors. One of the most significant genetic factors is human leukocyte antigen (HLA) polymorphism. Specifically, the HLA-DRB1*04 variant has been associated with the presence of rheumatoid nodules [8,9]. In terms of epigenetic factors, the presence of important enzymes like fat mass and obesity-associated protein (FTO), which modifies N6-methyladenosine (m6A) methylation and acts as a demethylase [10], has been described. However, in RA patients, its function is reduced, leading to higher levels of m6A in peripheral blood [11]. Moreover, authors have found a relationship between altered FTO levels, contents of another m6A demethylase enzyme called AlkB homolog 5 (ALKBH5) [12], the enzyme recognizing m6A modifications called YTH N6-methyladenosine RNA binding protein F2 (YTHFD2) [13], and RA activity [11].

Another factor that can contribute to RA is alterations in the gut microbiota, which can occur up to 5 years before disease onset [14]. Some of these variations include decreased *Pseudomonas*, *Ruminococcus*, or *Coprococcus*, while other bacterial species like *Lactobacillus*, *Raoultibacter*, or *Eubacterium* increase significantly when compared to healthy controls [14]. In addition, an association has been observed between the infection caused by the bacteria responsible for periodontitis, *Porphyromonas gingivalis*, and the onset of RA symptoms in a murine model of collagen-induced arthritis (CIA) [15].

One of the environmental or dietary factors that can also contribute to the onset of RA is smoking. Previous studies have demonstrated that continuous cigarette consumption for a minimum of 20 years promotes the development of this disease [16]. Additionally, alveolar macrophages isolated from smokers show nearly twice as much citrullination positivity compared to those in non-smokers [17]. Citrullination originates from a post-translational modification of arginine, mediated by the enzyme peptidylarginine deiminase enzyme (PAD) [18]. This phenomenon typically occurs under inflammatory conditions, and citrullinated proteins are recognized by autoantibodies called anti-citrullinated antibodies (ACPA), the presence of which in patients’ serum is considered a diagnostic marker for RA [18]. Another autoantibody used as a diagnostic marker for the disease is rheumatoid factor (RF), which recognizes the crystallizable fraction (Fc) of immunoglobulins (Ig) G [19]. Both ACPA and RF are present in the serum of patients who will develop this disease years later [20].

Many studies have analyzed the influence of different leukocyte populations and autoantibodies in disease onset and progression, for example, cytotoxic T lymphocytes, which are increased in RA patients’ blood [21], correlate with worsening patient symptoms [22,23] or variations in regulatory T cells, whose diminished levels are associated with exacerbated symptoms in patients at the moderate disease stage [24].

Other important leukocyte populations in the pathogenesis of RA are the phagocytic cells, such as neutrophils and macrophages. Neutrophils have been observed to be related to RA onset and development through mechanisms such as the degranulation of reactive oxygen species or the release of neutrophil extracellular traps (NETs) [25]. On the other hand, monocytes/macrophages are important in RA pathogenesis due to the release of pro-inflammatory cytokines. Previous studies have analyzed macrophage polarization into two populations: M1 and M2, with the former being pro-inflammatory and producing cytokines like interleukin (IL)-6 and tumor necrosis factor (TNF)-α, while the latter having anti-inflammatory function by secreting IL-10 and repairing damaged tissue [26]. RA patients with exacerbated symptoms have a higher number of M1-type macrophages [27]. The release of these pro-inflammatory cytokines, such as IL-1, IL-17A, IL-21, and IL-23 by macrophages and other immune cells creates a pro-inflammatory environment which is often correlated with markers like disease activity score (DAS) 28 and C reactive protein (CRP), as well as increased bone erosion [28,29,30,31,32].

As previously mentioned, synovitis is frequently found in the early stages of RA. This condition occurs when cells of the immune system migrate to this tissue, creating a pro-inflammatory environment. In cartilage, both chondrocytes and fibroblast-like synoviocytes (FLS) play a significant role in inflammation. Studies in mice with severe immunodeficiency (SCID) observed that the inhibition of chondrocytes suppressed the migration of FLS from patients to the cartilage matrix, and this migration could be restored by the addition of IL-1β [33].

T-helper (Th)1 lymphocytes also infiltrate the synovium and produce IL-17, which in turn increases the production of IL-1 and TNF-α, inducing cartilage destruction and inhibiting proteoglycan synthesis [34]. Additionally, neutrophils that infiltrate the synovium also have a detrimental effect on the cartilage by releasing matrix metalloproteinases (MMPs) [35]. Compared to synovial fluid from osteoarthritis patients, RA subjects have higher concentrations of MMP-1, MMP-2, MMP-3, MMP-8, and MMP-9, especially MMP-3 which reaches extremely elevated levels [36,37]. Moreover, chondrocytes release pro-inflammatory cytokines (IL-1, IL-17, IL-18, and TNF-α), promoting their catalytic function and degradation of the extracellular matrix [38]. One of the cytokines that plays a more significant role is IL-1β, as it promotes the expression of various types of MMPs, such as MMP-1 or MMP-9 [39,40]. In the most severe stages of RA, bone erosion occurs, involving osteoclasts, which will be discussed next.

### 1.2. The Osteoclasts

There are two types of cells responsible for maintaining bone balance: osteoblasts and osteoclasts. Osteoclasts are multinucleated cells derived from macrophages [41]. However, for proper osteoclast differentiation, two specific molecules are required: macrophage colony-stimulating factor (M-CSF) and receptor activator for nuclear factor κ B ligand (RANKL) [42]. Normally, osteoclasts reside in bone and play a crucial role in removing and generating new bone tissue under physiological conditions [43]. However, in the case of RA, these cells are found in the synovial membrane [44], where their primary function is bone degradation. Osteoclast precursors circulate in the bloodstream and are stimulated by TNF-α, which promotes osteoclast formation and, consequently, bone erosion [45]. Therefore, TNF-α acts synergistically with RANKL to enhance osteoclastogenesis [46]. In RA, there is an imbalance in bone formation/resorption due to an increase in osteoclast formation compared to osteoblasts. Thus, it is not surprising that patients with this disease commonly exhibit bone erosion. Osteoclasts attach to bone via the αVβ3 integrin [47] and facilitate bone degradation by creating an acidic environment through the release of H^+^ and Cl^−^ [47,48] and expressing carbonic anhydrase [49] as well as releasing tartrate-resistant acid phosphatase (TRAP), cathepsin K, and MMP-9 [47], which aid in bone calcium dissolution and the formation of resorption lacunae. Additionally, they also express vitronectin receptors [50] and calcitonin receptors [51].

RANKL is a member of the TNF receptor family, and aside from being present in macrophages and osteoblasts, it is also expressed in other cell types in RA. For instance, it is found in activated Th lymphocytes and FLS, inducing osteoclast formation [52]. Its expression in FLS is controlled by IL-17 through signal transducer and activator of transcription 3 (STAT3) activation [53]. The interaction between RANKL and its receptor RANK depends on the molecule osteoprotegerin (OPG) [54], a soluble member of the TNF receptor family. OPG binding to RANKL inhibits osteoclast formation and bone resorption function [55]. An important finding is that RANKL levels are elevated in the synovial fluid of RA patients, while OPG levels are reduced [52]. Although RANKL expression was initially believed to be limited to T lymphocytes, it has been observed that memory B lymphocytes in RA patients also spontaneously express it in higher amounts than T lymphocytes, promoting osteoclast formation and differentiation [56]. In terms of T lymphocytes, a study has revealed that senescent CD4^+^CD28^−^ Th cells express higher levels of RANKL than active CD4^+^CD28^+^ Th cells, contributing to bone erosion [57].

In RA, bone destruction is also related to the activity of specific cytokines. For example, it has been demonstrated in murine models that IL-1 absence protects against bone destruction [58]. Another cytokine that plays a significant role is IL-17 [34], while IL-4 has opposite effects and can help protect the bone [59]. An additional cytokine contributing to processes leading to bone erosion is IL-6. Previous studies have confirmed the important role of this cytokine in bone and cartilage destruction, as an inverse correlation between IL-6 levels and the T-score, an indicator of bone density, has been observed [60]. Other factors influencing bone resorption and remodeling include prostaglandin E2 (PGE2) [61], vitamin D [62], and parathyroid hormone [63].

Genetically, there is a gene that plays a significant role in bone erosion called Dickkopf-1 (*DKK-1*). This gene inhibits the wingless-related integration site (Wnt) pathway and positively regulates the β-catenin/T-cell factor pathway [64]. In patients with chronic inflammation like RA, an increase in *DKK-1* gene expression has been observed, along with reduced trabecular bone score (TBS) values (a bone density marker) [65]. This suggests that *DKK-1* gene promotes osteoclast formation and bone degeneration. Additionally, *DKK-1* gene expression and protein production are influenced by TNF-α and begin before the onset of the disease [66]. Another relevant protein is secreted frizzled-related protein-1 (sFRP1), which also inhibits the Wnt/β-catenin pathway and promotes the differentiation of memory T lymphocytes, which are producers of transforming growth factor (TGF)-β. In individuals with RA, there is an elevation in sFRP1 levels [67].

Osteoclasts also have a role in the degradation of the mineral layer of the cartilage [68]. However, in the proper maintenance of the cartilage, lubricin plays a crucial role. Lubricin is a glycoprotein encoded by the proteoglycan 4 (*PRG4*) gene. In murine models, the lack of this molecule has been observed to be associated with the absence of chondrocytes in the synovium and excessive hyperplasia of this tissue [69]. Additionally, fibroblast growth factor (FGF) also has beneficial effects on the joints. Studies have shown that in murine models lacking FGF-2, mice rapidly develop osteoarthritis with age or after knee surgery [70].

Another enzyme involved in cartilage degradation is a disintegrin and metalloproteinase with thrombospondin motifs (ADAMTS)4-5, which breaks down the present aggrecan in the joints. Inhibiting this enzyme in mice resulted in the preservation of cartilage with normal characteristics [71]. In proper homeostasis, there should be a balance between synthesis and degradation. Therefore, it is not surprising to find the expression of the natural inhibitor of these enzymes in the synovium, called tissue inhibitor of metalloproteinases (TIMP)1 [36], which is also produced by osteoblasts, osteoclasts, osteocytes, and chondrocytes [72]. However, in RA, this balance tilts toward cartilage degradation.

### 1.3. Exosomal miRNAs

Exosomes are small extracellular vesicles (EVs) initially discovered by Peter Wolf in 1967 as “platelet-dust” [73]. However, knowledge in this field has considerably advanced, and now we know that there are three types of EVs: microvesicles (100–1000 nm), apoptotic bodies (500–4000 nm), and exosomes, the latter being the smallest (30–150 nm) [74,75]. Exosomes have been found in a wide variety of tissues and fluids, such as blood [76], serum [77], urine [78], and feces [79], among others. Furthermore, it has been demonstrated that tumor cells [80], bacteria like *Escherichia coli* [81], or mesenchymal stem cells (MSCs) [82] are capable of producing and releasing exosomes.

The process of exosome formation begins with the synthesis of intraluminal vesicles (ILVs), involving the endosomal sorting complexes required for transport (ESCRT) [74]. These complexes perform various functions, such as inducing membrane deformations and separating newly formed vesicles [83,84]. ILVs are formed within late endosomes known as multivesicular bodies (MVBs) [85]. However, it is not until MVBs fuse with the plasma membrane and release their content into the extracellular environment that ILVs are referred to as exosomes [85]. Sometimes, instead of releasing their content, MVBs degrade ILVs through fusion with lysosomes [86].

Exosomes present a wide variety of markers on their surface. These markers include heat shock proteins (HSPs) [87], class I major histocompatibility complex (MHC) [88], annexins [87], guanosine triphosphatases (GTPases) [89], and tetraspanins CD9, CD81, and CD63, which are especially known for their utility in detecting and characterizing these extracellular vesicles [90].

As for the internal content of exosomes, it includes proteins, RNA, and DNA molecules. Regarding RNA molecules, mRNA is found, although microRNAs (miRNAs) play a more relevant role due to their capacity to regulate gene expression. Furthermore, exosomes contain RNA-binding proteins (RBPs) [91]. Notably, the case of argonaut 2 (AGO2) is significant, as there is controversy among researchers about whether this protein is found within exosomes, stabilizing miRNAs, or if it is only present in the extracellular environment [92,93]. As for DNA, the presence of genomic [94] and mitochondrial [95] material fragments has been found.

Regarding miRNAs, their structure consists on single-stranded RNA molecules composed of approximately 20 nucleotides [96], and their main function is to regulate gene expression [97]. Their synthesis begins in the cell nucleus, where a RNA polymerase II transcribes specific genes to form primary or pri-miRNAs [98]. Next, the protein DiGeorge syndrome critical region 8 (DGCR8) facilitates the binding of the enzyme Drosha, which processes pri-miRNA to form pre-miRNA, which is approximately 70 nucleotides long [99,100,101]. The pre-miRNA is then transported to the cytoplasm by exportin 5 [102], where the complex composed of Dicer and transactivation response element RNA-binding protein (TRBP) processes it to form an intermediate duplex [103,104]. One of the strands of the miRNA, called passenger or miRNA*, will be degraded, while the other strand will bind to the RNA-induced silencing complex (RISC) [105], interacting with the protein AGO [106,107] and carrying out its gene silencing function [108].

## 2. miRNAs Released by Osteoclasts

The study of miRNAs in RA has focused on their potential utility as diagnostic markers (including let-7d-5p, miR-24-3p, miR-126-3p, miR-130a-3p, miR-221-3p, and miR-431-3p), indicators of disease progression (such as miR-22, miR-486-3p, and miR-382), or markers associated with bone erosion (like miR-99b-5p). This analysis typically involves assessing these miRNAs in synovial fluid or serum [109,110,111]. However, few articles have delved into the specific function of osteoclasts in bone erosion through the release of exosomal miRNAs. For this reason, we have reviewed the miRNAs contained in exosomes released by osteoclasts and grouped them according to their ability to promote or inhibit osteoblast and osteoclast differentiation (Figure 1).

### 2.1. Exosomal miRNAs with the Ability to Promote Osteoblast Differentiation

Exosomes isolated from osteoclasts derived from bone marrow macrophages of C57 black 6 (C57BL/6) mice contain miR-18a-5p, miR-185-5p, miR-142-3p, miR-106a-5p, and miR-132-3p, but only miR-106a-5p has the ability to promote bone formation and differentiate bone mesenchymal stem cells through its action on family with sequence similarity 134, member A (*FAM134A*), a gene related to tumor proliferation, metastasis, etc. (Figure 2) [112,113]. Additionally, miR-142-3p increases alkaline phosphatase enzymatic activity in bone mesenchymal stem cells (Figure 2) [113], an enzyme produced by osteoblasts and used as a marker of their activity [114,115]. Furthermore, a study conducted under microgravity conditions with the murine macrophage cell line Ralph, Raschke, Watson (RAW) 264.7, differentiated into osteoclasts with RANKL, concluded that miR-142a-3p decreased its expression in the extracellular vesicles of osteoclasts [116]. In the same study, a decrease in the expression of miR-34a, miR-96-5p, miR-199a, and miR-20a was observed, while miR-486b-5p showed higher expression in extracellular vesicles released by osteoclasts (Figure 2) [116]. In all cases, these miRNAs have been associated with the ability to promote osteoblast differentiation, with miR-96 acting through the Wnt pathway and miR-20a acting through the bone morphogenetic protein (BMP)/runt-related transcription factor 2 (RUNX2) pathway [117,118,119,120,121,122]. Another miRNA capable of promoting osteoblast differentiation is miR-324 (Figure 2). Osteoclasts differentiated from the bone marrow macrophages of mice released extracellular vesicles with high miR-324 expression, which exhibited osteogenic capacity through the regulation of the Ras homology GTPase-activating protein 1 (ARHGAP1)/Ras homolog family member A (RhoA)/Rho-associated coiled-coil containing protein kinase (ROCK) signaling pathway [123]. This led to increased mineralization, regeneration, and bone density.

### 2.2. Exosomal miRNAs with the Ability to Inhibit Osteoblast Differentiation

The group led by Yang analyzed exosomes released by the RAW 264.7 cell line after differentiation into osteoclasts. The results of their research showed that exosomes from osteoclasts contain miR-23a-5p, with an inhibitory function on osteoblasts (Figure 3). Furthermore, using bioinformatics techniques, these authors demonstrated that the osteogenic effect of miR-23a-5p could be due to the inhibition of RUNX2 transcription factor. As suggested in the article, this inhibition could be mediated by the yes-associated protein 1 (YAP-1) transcription factor, which interacts with RUNX2 and inhibits metallothionein 1D, pseudogene (MT1DP) [124], a long non-coding RNA (lncRNA) that promotes cytotoxic function in cells [125].

In another study, the same cell line used previously was differentiated into osteoclasts with RANKL and, after analyzing the released exosomes, high concentrations of miR-214 were observed (Figure 3) [126,127]. Moreover, these exosomes could be internalized by osteoblasts through the ephrin A2/ephrin A2 receptor (EphA2) system, inhibiting the function of these cells [127]. The expression of this miRNA was also observed to be increased in extracellular vesicles released by osteoclasts under microgravity conditions, while miR-29b levels were decreased (Figure 3) [116]. Both miRNAs act as inhibitors of osteoblast differentiation, and in the case of miR-29b, it exerts its function by reducing insulin-like growth factor 1 (IGF-1) levels, as observed in an ex vivo model with murine osteocyte cells stimulated by mechanical stretching [128,129].

### 2.3. Exosomal miRNAs with the Ability to Inhibit Osteoclast Differentiation

In other studies conducted with osteoclasts differentiated from mouse bone marrow precursors, miR-146a was found to be overexpressed in the extracellular vesicles of osteoclasts (Figure 4) [126,130]. Studies carried out with osteoclasts differentiated from monocytes from the peripheral blood mononuclear cells of patients with Paget’s syndrome showed decreased levels of miR-146a, leading to inhibition of osteoclasts and bone resorption, mediated by the nuclear factor—κB (NF-κB) transcription factor [131].

The results of miRNA expression contained in extracellular vesicles released by osteoclasts under microgravity conditions concluded that miR-22-3p, miR-26a-5p, miR-27a-3p, miR-29a-3p, and miR-125b-5p showed decreased levels (Figure 4) [116]. All these miRNAs play a crucial role as inhibitors of both osteoclasts and osteoclastogenesis, with miR-29a, for instance, acting as an inhibitor of RANKL, which can be regulated due to estrogen deficiency, and miR-26a acting through the connective tissue growth factor (CTGF) pathway [132,133,134,135,136].

## 3. Possible Implications in Patients with RA

While the exosomal miRNAs previously discussed have been studied in murine models, their results could lay the groundwork for potential therapeutic targets in RA patients, aiming to decrease osteoclastogenesis and bone erosion. For this reason, the use of techniques that inhibit or overexpress specific miRNAs, depending on the function we want to enhance, could be a future therapeutic strategy. However, the effects of the exosomal miRNAs presented in this review on osteoclasts have not yet been analyzed. Therefore, we will present the outcomes of their inhibition or overexpression in various cells in the context of RA.

One way to inhibit exosomal miRNAs is by using antagomiRs, molecules complementary to miRNAs that block their function, such as with miR-21 and its antagomiR [137]. In the case of this study, we would be interested in inhibiting the function of miR-23a, miR-29b, and miR-214, as they have a negative effect on osteoblast proliferation. However, the inhibition of all these miRNAs in the context of RA has not been studied, so we will focus on miR-23a and miR-214. The inhibition of miR-23a leads to an increase in the nuclear factor of activated T cells 1 (NFATc1), MMP-9, cathepsin K, and TRAP, while decreasing TGF-β-activated kinase 1 (TAK1). Additionally, it promotes osteoclast generation [138], increases the release of pro-inflammatory cytokines (TNF-α, IL-6, and IL-23), TLR4, and TNF receptor associated factor (TRAF)2 in RAW 264.7 macrophages and synovial cells [139]. On the other hand, the use of antagomiR-214 results in a decrease in MMP-3, MMP-13, and chondrocyte apoptosis [140].

Another strategy to inhibit miRNA function would be the use of blockmiRs, which are sequences complementary to the target mRNA. This approach achieves more precise inhibition without side effects compared to antagomiRs. This technique has not been employed for the miRNAs included in this RA study, but it has been used for diabetic retinopathy, where blockmiRs prevented miR-27a from reducing vascular endothelial (VE)-cadherin transcription [141]. In mice with induced diabetic retinopathy treated with blockmir CD5-2, fewer lesions, reduced vascular permeability, and less microglia activation were observed [141]. This technique has also been used in a mouse model of unilateral hindlimb ischemia, inhibiting thrombin permeability and vascular endothelial growth factor (VEGF)-associated vascular permeability, thus promoting recovery after this condition [142]. This is an innovative technique that requires further research in RA.

On the other hand, in cases where miRNAs are overexpressed, agomiRs or the mimics technique is used. For example, to promote osteoblast proliferation or inhibit osteoclasts, we could use the agomiR or mimics technique to overexpress miR-22, miR-26a, miR-27a, miR-29a, miR-125b, miR-146a, miR-20a, miR-34a, miR-96, miR-142, miR-199a, miR-324a, and miR-486b. The effect of overexpressing all these miRNAs in the context of RA has not been studied, so we will focus on miR-22, miR-26a, miR-27, miR-20a, miR-34a, miR-142 and miR-146a.

The overexpression of miR-22 increases apoptosis and decreases IL-6 release in a human rheumatoid fibroblast-like synoviocyte cell line (MH7A cells), inhibiting synovial inflammation [143]. However, in another study, overexpression of this miRNA produced contrasting results, with decreased cell inhibition rate, apoptosis, and increased release of pro-inflammatory cytokines (IL-1β, IL-6, IL-8), PGE2, MMP-3, or inducible nitric oxide synthase (iNOS) [144].

Conversely, the increased expression of miR-26a reduces joint swelling, arthritis score, joint damage, chondrocyte apoptosis, release of pro-inflammatory cytokines (IL-1β, TNF-α, IL-6), and CTGF protein expression in the murine model of RA [145] and Tohoku Hospital Pediatrics-1 (THP-1) cells [146]. Additionally, the overexpression of this miRNA reduces TNF-α expression, as well as clinical symptoms of synovitis in a pristine-induced arthritis (PIA) rat model [146].

Exploring additional miRNAs, the use of agomiR-27 promotes FLS proliferation and IL-8 expression [147], while agomiR-34a decreases FLS proliferation, TNF-α, MMP-1, MMP-9, and IL-6 expression, and retains the cells in G1 phase of the cell cycle [148]. The overexpression of miR-20a using agomiRs reduces osteoclast proliferation and differentiation, as well as their erosive function [149]. It also reduces IL-6 release in RA-FLS and TNF-α and IL-1β in THP-1 cell-derived macrophages [150].

Finally, the overexpression of miR-142 and miR-146a in B lymphocytes increases age-associated B cells (ABC) [151]. The use of agomiR-146a reduces extracellular matrix metalloproteinase inducer (EMMPRIN), VEGF, and MMP-9 secretion in the HT1080 fibroblast cell line [152]. In RA-FLS the overexpression of this miRNA reduces cell density, apoptosis, and proliferation, while increasing cell inhibition rates leading to a decrease in arthritis score [153]. Finally, the administration of liposomes with agomiR-146a in the CIA mice model reduces the presence of osteoclasts and the erosion score [154].

Another strategy to rebalance the presence and the function of osteoclasts and osteoblasts would be the use of competitive endogenous RNAs (ceRNAs), which are molecules with binding sites for miRNAs, blocking their function. Among ceRNAs are pseudogenes, lncRNAs, or circular RNAs (circRNAs), such as phosphatase and tensin homolog pseudogene 1 (*PTENP1*), X inactive specific transcript (XIST), or circular RNA sponge for miR-7 (ciRS-7) [155,156,157]. The function of ceRNAs has been studied in diseases like RA, where molecules like protein tyrosine phosphatase non-receptor type 22 (*PTPN22*) [158], nuclear paraspeckle assembly transcript 1 (*NEAT1*) [159,160], P38 inhibited cutaneous squamous cell carcinoma associated lincRNA (PICSAR) [161], or circular RNA nucleoporin 214 (circNUP214) [162] can modulate miRNA expression. In fact, these interactions can regulate the function of certain genes, either improving or exacerbating RA symptoms as seen in Table 1. Regarding the miRNAs of interest in this review, only the effect of lncRNA TRAF-type zinc finger domain containing (TRAFD) on miR-27a-3p has been studied in ex vivo models of RA [163]. Inhibition of this miRNA promotes the overexpression of C-X-C motif chemokine ligand (*CXCL1*), ultimately resulting in the inhibition of chondrocyte proliferation and migration. Thus, the TRAFD1-4:1/miR-27a-3p/*CXCL1* axis has been established, wherein the action of an lncRNA (TRAFD1-4:1) modulates the activity of a gene (*CXCL1*), producing beneficial effects in RA.

Nevertheless, these results are preliminary, and although targeting the miRNAs produced by osteoclasts could have beneficial effects for RA patients, further studies are needed to apply antagomir, blockmiRs, agomiRs or ceRNA technology in humans.

## 4. Conclusions

In the more advanced stages of RA, bone erosion mediated by osteoclasts occurs. Additionally, these cells can communicate with other osteoclasts or osteoblasts by releasing exosomal miRNAs, either promoting or inhibiting their proliferation and differentiation. To date, this is the first study that reviews the potential application of exosomal miRNAs produced by osteoclasts as markers of bone erosion or disease severity, aiding in the therapeutic management of RA and preventing patients from reaching the more severe stages that involve bone erosion. Moreover, the application of antagomiRs, blockmiRs, agomiRs, or ceRNAs to rebalance the proportion between osteoclasts and osteoblasts in order to block osteoclastic activity and promote osteoblastogenesis would be a therapeutic alternative for patients resistant to DMARDs or in very advanced stages of their disease. Specifically, we propose that the inhibition of miR-23a, miR-29b, and miR-214 would favor the proliferation of osteoblasts, while the overexpression of miR-22-3p, miR-26a, miR-27a, miR-29a, miR-125b, and miR-146a would propitiate the inhibition of osteoclasts. Additionally, increasing the expression of miR-20a, miR-34a, miR-96, miR-106a, miR-142, miR-199a, miR-324, and miR-486b would enhance the proliferation and differentiation of osteoblasts.

The challenges associated with utilizing miRNAs in RA as indicators of disease severity stem from a limited number of studies demonstrating the impact of exosomal miRNAs. Additionally, the potential therapeutic use of these molecules to impede bone erosion in patients faces various constraints. These include ensuring their effective delivery to the target tissue, understanding drug interactions, and considering patient-specific factors such as genetic and epigenetic variations, disease status, and co-morbidities [175]. Given the diverse effects that can emerge from targeting miRNAs specifically, further comprehensive research is necessary to precisely define the inhibition or overexpression of exosomal miRNAs. This is crucial for ensuring that the observed effects genuinely result from the actions exerted on these molecules. Furthermore, since the inhibition or overexpression of miRNAs has not been tested directly in osteoclasts but rather in other ex vivo or murine models, it is imperative to evaluate how the use of antagomiRs, blockmiRs, agomiRs, or ceRNAs responds in humans and what potential side effects this therapy may present for future applications in personalized medicine.

## Figures and Tables

**Figure 1 ijms-25-01506-f001:**
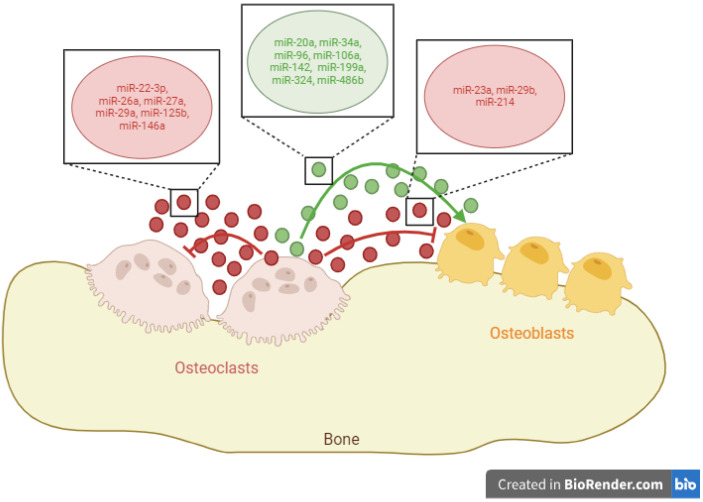
Exosomal microRNAs (miRNAs) released by osteoclasts and their functions over osteoblasts and other osteoclasts. Green circles/lines indicate exosomal miRNAs that promote cellular differentiation, while red circles/lines show exosomal miRNAs that inhibit cellular differentiation. Created with BioRender.com.

**Figure 2 ijms-25-01506-f002:**
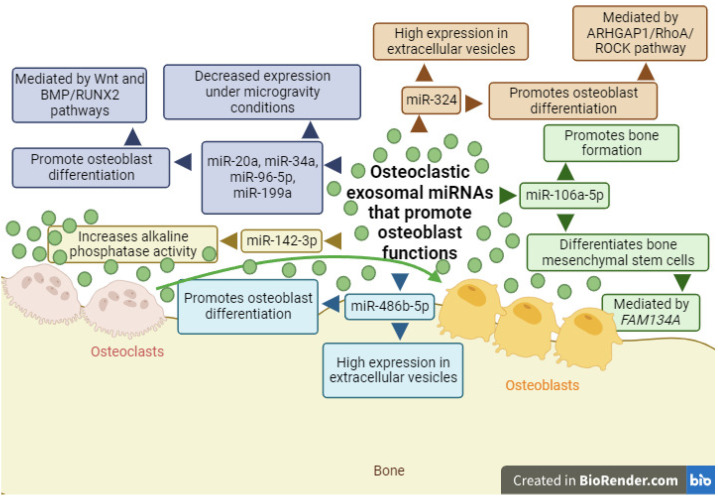
Diagram of exosomal miRNAs produced by osteoclasts (miR-20a, miR-34a, miR-96-5p, miR-106a, miR-142-3p, miR-199a, miR-324, and miR-486b-5p) and their function as enhancers of osteoblast functions. Green circles/lines indicate exosomal miRNAs that promote cellular differentiation. Wingless-related integration site (Wnt); bone morphogenetic protein (BMP)/runt-related transcription factor 2 (RUNX2); Ras homology GTPase-activating protein 1 (ARHGAP1)/Ras homolog family member A (RhoA)/Rho-associated coiled-coil containing protein kinase (ROCK); family with sequence similarity 134, member A (FAM134A). Created with BioRender.com.

**Figure 3 ijms-25-01506-f003:**
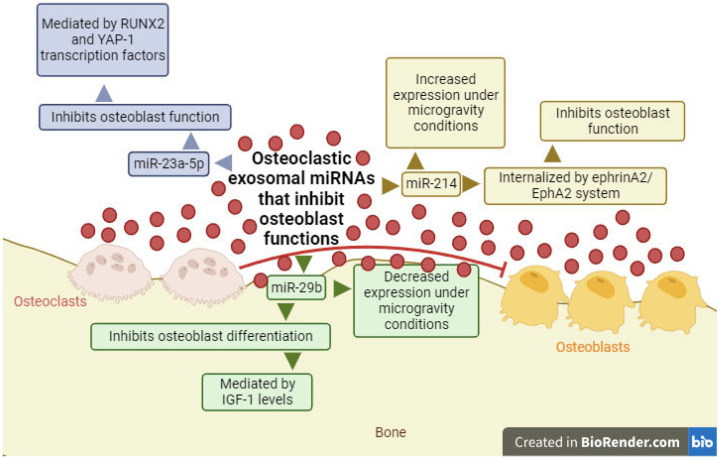
Diagram of exosomal miRNAs produced by osteoclasts (miR-23a-5p, miR29b, and miR-214) and their function as inhibitors of osteoblast functions. Red circles/lines show exosomal miRNAs that inhibit cellular differentiation. Yes-associated protein 1 (YAP-1); ephrin A2 receptor (EphA2); insulin-like growth factor 1 (IGF-1). Created with BioRender.com.

**Figure 4 ijms-25-01506-f004:**
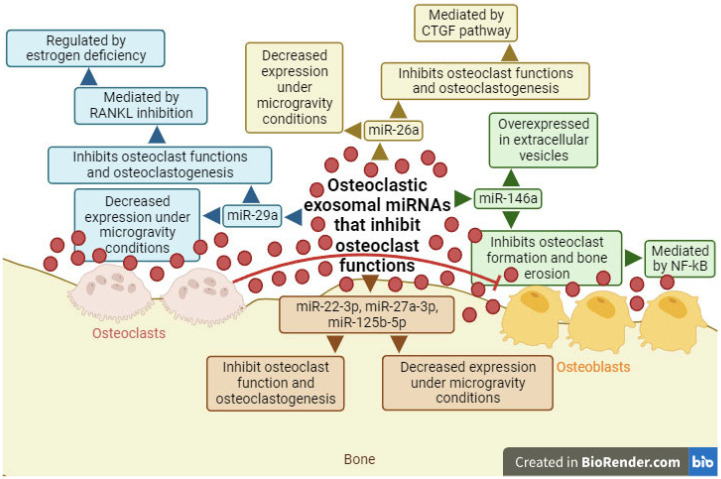
Diagram of exosomal miRNAs produced by osteoclasts (miR-22-3p, miR-26a, miR-27a-3p, miR-29a, miR-125b-5p, and miR-146a) and their function as inhibitors of others osteoclast functions. Red circles/lines show exosomal miRNAs that inhibit cellular differentiation. Receptor activator for nuclear factor κ B ligand (RANKL); connective tissue growth factor (CTGF); nuclear factor—κB (NF-κB). Created with BioRender.com.

**Table 1 ijms-25-01506-t001:** Relation of the different ceRNAs, their target miRNAs, the different molecules that are regulated and the effects in murine or culture models of RA. Maternally Expressed 3 (MEG3); protein kinase B (AKT); mechanistic target of rapamycin kinase (mTOR); growth arrest specific (GAS); sirtuin (Sirt); TNF receptor associated factor (TRAF) -type zinc finger domain containing (TRAFD); C-X-C motif chemokine ligand (CXCL); Kruppel-like factors transcription factor (KLF); family with sequence similarity 46 member C (FAM46C); oxidative stress responsive serine rich 1 antisense RNA 1 (OSER1-AS1); E2F transcription factor 1 (E2F1); opa interacting protein 5 antisense RNA 1 (OIP1-AS1); Wnt family member 7B (Wnt7b); activating transcription factor (ATF); Toll-like receptor (TLR); cyclin dependent kinase (CDK); pyruvate dehydrogenase kinase (PDK).

Axis	ceRNA	miRNA	Molecular Target	Effects on RA	Reference
TRAFD1-4:1/miR-27a-3p/*CXCL1*	TRAF1-4:1	miR-27a-3p	*CXCL1*	TRAF1-4:1 inhibited the proliferation and migration of chondrocytes	[163]
GAS5/miR-222-3p/Sirt1	GAS5	miR-222-3p	*SIRT1*	GAS5 regulates FLS function by inhibiting their proliferation and inflammation and promoting their apoptosis	[164]
circ_0130438/miR-130a-3p/KLF9	circ_0130438	miR-130a-3p	KLF9	circ_0130438 inhibits TNF-α-induced invasion, proliferation, migration and inflammation in MH7A cells	[165]
circRNA_17725/miR-4668-5p/FAM46C	circRNA_17725	miR-4668-5p	FAM46C	circRNA_17725 induces macrophage proliferation at M2	[166]
OSER1-AS1/miR-1298-5p/E2F1	OSER1-AS1	miR-1298-5p	E2F1	OSER1-AS1 inhibits apoptosis and inflammation of the RA-FLS	[167]
OIP5-AS1/miR-410-3p/Wnt7b	OIP5-AS1	miR-410-3p	Wnt7b	OIP5-AS1 inhibits FLS proliferation	[168]
hsa_circ_0001859/miR-204/211/ATF2	hsa_circ_0001859	miR-204/211	ATF2	Silencing of hsa_circ_0001859 reduces inflammation in SW982 cells	[169]
circRNA_09505/miR-6089/AKT1/NF-κB	circRNA_09505	miR-6089	AKT1/NF-κB	circRNA_09505 promotes the release of pro-inflammatory cytokines (TNF-α, IL-6, IL-8, IL-12 and IL-1β) from macrophages	[170]
HIX003209/miR-6089/TLR4	HIX003209	miR-6089	TLR4	HIX003209 promotes pro-inflammatory cytokine release (TNF-α, IL-6 and IL-1β) in macrophages	[171]
circ0003353/miR-31-5p/CDK1	circ0003353	miR-31-5p	CDK1	Overexpression of circ0003353 promotes proliferation, cell cycle progression and inflammatory cytokine production (IL-1β and IL-6) of RA-FLS	[172]
LOC100912373/miR-17-5p/PDK1	LOC100912373	miR-17-5p	PDK1	LOC100912373 induces FLS proliferation	[173]
lncRNAS56464.1/miR-152-3p/Wnt pathway	lncRNAS56464.1	miR-152-3p	Wnt pathway	lncRNAS56464.1 promotes FLS proliferation	[174]

## Data Availability

No new data were created or analyzed in this study. Data sharing is not applicable to this article.

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
