# Peer review of "Exosomal Osteoclast-Derived miRNA in Rheumatoid Arthritis: From Their Pathogenesis in Bone Erosion to New Therapeutic Approaches"

_ijms, 2024, doi:10.3390/ijms25031506_

Round 1

Reviewer 1 Report

Comments and Suggestions for Authors

This is quite an instructive review of exosomal osteoclast-derived miRNA in rheumatoid arthritis (RA) and their various roles in the pathogenesis of bone erosions in this disease. The authors have also described the potential uses of the miRNA system as a means of therapeutically targeting the inflammatory process in RA.

The exosomal miRNAs have been classified into three categories based on their different functions, namely osteoclast inhibitors, osteoblast inhibitors and osteoblast enhancers. A comprehensive list of molecules falling into these 3 categories is included and the targeting of these exosomal miRNAs, by the use of antagomiRs, blockmiRs, agomiRs and competitive endogenous RNAs (ceRNAs) is also discussed.

Some promising results are presented, mostly from murine systems, but these results are preliminary in nature and much further work in RA systems will be necessary to determine the overall effects of overexpressing or blocking these miRNAs, in the context of RA. Some conflicting data, depending on systems used, have also been reported e,g., miR-22  decreases IL-6 release in the human rheumatoid FLS cell line (MH7A cells) with reduced levels of synovial inflammation. However, overexpression of this same miRNA produced contrasting results, in a rat model of RA. 

General comments – 

The report has a good background section and a useful Figure and Table, as well as an extensive reference section. For completeness suggest including some articles that compare miRNA expression between healthy individuals and those that develop RA (PMID: 31273939 and PMID: 29273071)

Specific comments – 

 The review is well written with a few minor errors noted e,g., in Line 354 – suggest changing “contrary” to “contrasting”. 

in Line 374 – suggest changing “cell inhibition rate” to “cell inhibition rates”

Comments on the Quality of English Language

The quality of English is good - it is very well written

Reviewer 2 Report

Comments and Suggestions for Authors

The paper is well-structured, with clear headings and subheadings that guide the reader through complex content. The flow from general RA discussion to specific miRNA details is logical and easy to follow. 

  1. However, the paper would be more balanced if the author could highlight the current research limits and the specific challenges in studying miRNAs in rheumatoid arthritis.
  2. The author might improve the paper by including additional diagrams or charts that show how miRNAs interact in rheumatoid arthritis, making it easier to understand.
  3. The author should emphasize what makes this review unique compared to others. They should point out any new ideas, viewpoints, or findings that set this paper apart.
